# Anti-Candidal Marine Natural Products: A Review

**DOI:** 10.3390/jof9080800

**Published:** 2023-07-28

**Authors:** Arumugam Ganeshkumar, Juliana Caparroz Gonçale, Rajendran Rajaram, Juliana Campos Junqueira

**Affiliations:** 1Department of Biosciences and Oral Diagnosis, Institute of Science and Technology, Sao Paulo State University (UNESP), Sao Jose dos Campos 12245-000, Brazil; juliana.goncale@unesp.br; 2Department of Materials Physics, Saveetha School of Engineering, Saveetha Nagar, Thandalam, Chennai 602105, India; 3Department of Marine Science, Bharathidasan University, Tiruchirappalli 620024, India; drrajaram69@rediffmail.com

**Keywords:** marine natural products, structural elucidation, antifungal properties

## Abstract

*Candida* spp. are common opportunistic microorganisms in the human body and can cause mucosal, cutaneous, and systemic infections, mainly in individuals with weakened immune systems. *Candida albicans* is the most isolated and pathogenic species; however, multi-drug-resistant yeasts like *Candida auris* have recently been found in many different regions of the world. The increasing development of resistance to common antifungals by *Candida* species limits the therapeutic options. In light of this, the present review attempts to discuss the significance of marine natural products in controlling the proliferation and metabolism of *C. albicans* and non-*albicans* species. Natural compounds produced by sponges, algae, sea cucumber, bacteria, fungi, and other marine organisms have been the subject of numerous studies since the 1980s, with the discovery of several products with different chemical frameworks that can inhibit *Candida* spp., including antifungal drug-resistant strains. Sponges fall under the topmost category when compared to all other organisms investigated. Terpenoids, sterols, and alkaloids from this group exhibit a wide array of inhibitory activity against different *Candida* species. Especially, hippolide J, a pair of enantiomeric sesterterpenoids isolated from the marine sponge *Hippospongia lachne*, exhibited strong activity against *Candida albicans*, *Candida parapsilosis*, and *Candida glabrata*. In addition, a comprehensive analysis was performed to unveil the mechanisms of action and synergistic activity of marine products with conventional antifungals. In general, the results of this review show that the majority of chemicals derived from the marine environment are able to control particular functions of microorganisms belonging to the *Candida* genus, which can provide insights into designing new anti-candidal therapies.

## 1. Introduction

Invasive and chronic fungal infections need particular medical attention since they are associated with a significant rise in treatment costs and a high fatality rate [1]. Patients who have lengthy hospital stays with compromised immune systems are more likely to get fungal infections. *Candida*, *Aspergillus*, and *Cryptococcus* are the three main genera of fungi that cause human diseases, ranging from superficial to invasive infections [2,3]. In contrast to other causes, they account for 90% of fungal infections that collectively afflict more than a billion people worldwide [4,5]. The four principal groups of antifungal drugs are polyenes, azoles, echinocandins, and pyrimidine analogues, which have been extensively employed in recent decades [6,7]. The therapeutic effectiveness of these antifungals depends on several factors, including host immunological response, fungal isolate origin, antifungal drug characteristics, and the emergence of drug tolerance and drug resistance. Among these factors, drug resistance is the prime concern in the world population since microorganisms can acquire several defense mechanisms against different classes of drugs [8,9]. These mechanisms can suppress the action of drugs by reducing the binding affinity with the drug target in fungal cells, leading to the overexpression or mutation in drug targets, the overproduction of hydrolytic enzymes, and alteration in effective drug concentration through the modulation of efflux activity [9,10]. 

Natural products are chemical molecules produced by several organisms living in different habitats [11,12]. There is a high degree of chemical diversity among biological organisms, resulting in unique structural and functional properties. Natural products have crucial roles in cellular processes, and many of them have been correlated with important biological functions [13]. The importance of natural products has been extensively reported in research focused on developing novel drugs against life-threatening conditions. The structures of natural products are entirely different from synthetic chemical products; they are more complex and have specific biological properties. The differentiation of natural products from synthetic chemical libraries provides a potential source for the identification of newer chemicals. The identification of natural products has been established since the 19th century and comprises chemical structures from terrestrial and marine environments; however, studies associated with marine natural products are limited compared with those focused on terrestrial sources.

Around 70% of Earth’s space is occupied by oceans, enriched with different floral and faunal diversity. To adapt to the harsh environment, marine organisms evolved themselves, thus directly or indirectly incorporating valuable chemical compounds with unique properties. For example, more than 30,000 chemical compounds were reported from the marine environment with anticancer properties [14,15]. Nevertheless, the extrapolation of marine compounds from the deep sea is still challenging due to many factors, including high cost, the requirement of highly sensitive instruments, time consumption, and the workforce employed [15]. Moreover, the quantity of compounds produced by an organism is relatively small, and chemical synthesis is hampered by its complex structural features. Currently, total synthesis and semi-synthesis are commonly used to overcome the supply–demand challenges of natural products. For this, biotechnological approaches are encouraged to obtain the maximum level of specific compounds using some large-scale fermentation techniques [16,17]. 

During the past few decades (1965–2021), only 17 of the purified chemical constituents from marine environments were recognized by US FDA for the prophylaxis of simple to life-threatening clinical conditions (Figure 1). Recently (between 2020 and 2021), lurbinectedin (ZepzelcaTM), belantamab mafodotin-blmf (Blenrep™), disitamab vedotin (Aidixi™), and tisotumab vedotin-tftv (TIVDAK™) obtained from tunicate and mollusk/cyanobacterium were approved for differential treatment of most common cancers, including metastatic cervical cancer, metastatic small cell lung cancer, and multiple myeloma [18]. In addition, there are 6 compounds in Phase III (originating from fungi, bacteria, puffer fish, tunicate, mollusk, and cyanobacteria), 15 in Phase II (originating from mollusk, cyanobacteria, sponge, and bryozoan), and 16 in Phase I (originating from mollusk, cyanobacteria, sponge, and red algae) stages of clinical trials. Besides the anticancer activity of the listed compounds, few other compounds have proven their effectiveness against chronic pain (tetrodotoxin) [19], coronavirus (plitidepsin) [20], relapsed or refractory systemic amyloidosis (STI-6129) [21], Alzheimer’s disease (bryostatin) [22], and HIV prevention and COVID-19 prophylaxis [23]. 

In relation to antifungal action, there are several review publications available about natural compounds from different sources; nonetheless, there are no specific publications on anti-candidal metabolites from marine resources. In light of the aforementioned antifungal resistance concerns, we were prompted to seek detailed information on marine natural products that can be effective against *C. albicans* and some non-*albicans* species. We expect that this review, which encompasses more than 150 articles from the previous three decades, can fill the knowledge gap regarding natural products targeting *Candida* infections. In this review, marine natural products from sponges, algae, sea cucumber, bacteria, fungi, and other organisms are presented and discussed in relation to their specific properties against *Candida* spp. (Figure 2).

## 2. Marine Natural Products

### 2.1. Sponges

A notable benthic community that may be found in many habitats of fresh and marine water is the sponge [24]. The generation of bioactive compounds by sponges has been closely linked to the enrichment of the sponge community with distinct populations of bacteria. Like other filter feeders, sponges never move about their surroundings in search of food and cannot flee from predators. Meanwhile, all sponge species continually produce specific substances as a defense strategy against their predators, including fish, turtles, and invertebrates [25,26]. The biological activities of these substances have been explored, and many of them showed activity against *Candida* spp., which are presented in Figure 3 and Table 1 and discussed below.

#### 2.1.1. Glycoside Derivatives

Among glycoside derivatives, two compounds from the marine sponge *Oceanapia* sp. have been investigated: oceanalin A (**C38**) and B (**C2**). These compounds are sphingoid tetrahydoisoquinoline β-glycosides unexpectedly discovered in the organic extract from this sponge. Previous studies proved their in vitro antifungal action against *C. glabrata*, in which oceanalin A (**C38**) showed a minimum inhibitory concentration (MIC) of 30 µg/mL [43], and oceanalin B (**C2**) exhibited a MIC of 25 µg/mL [28]. Oceanapiside (**C1**), another compound purified from the methanol extract of the sponge *Oceanapia phillipensis,* also showed activity against *C. glabrata.* This compound was tested on a fluconazole-resistant strain, and its mechanism of action was associated with a disturbance in the sphingolipid pathway [27]. However, oceanapiside (**C1**) was not active against *C. albicans* and *C. krusei* strains [54].

In addition, tetramic acid glycoside compounds, called aurantosides, have been studied as potential antifungal agents. Aurantosides D (**C46**), E (**C47**), and F (**C48**) were isolated from the marine sponge *Siliquariaspongia japonica*. Among them, only aurantosides D (**C46**) and E (**C47**) were found to be active against *C. albicans*, with inhibition zones of 9.5 and 9.7 mm and MICs of 11 and 13.6 μg/mL, respectively [47]. Aurantoside K (**C23**) was isolated from the Fijian marine sponge *Melophlus* and showed a wide spectrum of antifungal activity against drug-resistant *C. albicans* strains, with MICs of 31.25 μg/mL and 1.95 μg/mL [39]. Aurantoside J (**C32**), another tetramic acid glycoside isolated from an Indonesian specimen of *Theonella swinhoei*, was found to be active against all the *Candida* strains tested (MIC >16 µg/mL), including *C. albicans*, *C. parapsilosis*, *C. glabrata*, and *C. tropicalis* [41].

#### 2.1.2. Alkaloids

Several alkaloid compounds from marine sponges with anti-candidal activity were reported, such as hemimycalins, nakamurines, agelasines, nagelamides, zamamidine, and ceratinadins. Hemimycalins A (**C13**) and B (**C14**) are newly discovered hydantoin alkaloids from *Hemimycale arabica*, a marine sponge found in the Red Sea. Both of these alkaloids showed activity against *Escherichia coli* and *C. albicans* at 100 μg/disc, resulting in inhibition zones of 10–20 mm [33].

Nakamurines A (**C10**) and B (**C11**) are new non-brominated pyrrole alkaloids isolated from the sponge of *Agelas nakamurai* that exhibited antifungal activity against *C. albicans*, with a MIC of 60 μg/mL found for nakamurine B (**C11**) [32]. There are other relevant compounds extracted from *Agelas* sp., including bromopyrrole alkaloids, which demonstrated antifungal activity against *C. albicans* in a *Caenorhabditis elegans* model of candidiasis [55]; agelasines O-U (**C24–C30**) (diterpene alkaloids) with activity against *C. albicans*, exhibiting MIC values of 16–32 μg/mL [31]; nagelamides U (**C18**) and W (**C19**), which showed antifungal activity on *C. albicans*, with IC50 values of 4 μg/mL [37]; and nagelamides X-Z (**C20–C22**), which demonstrated strong activity against *C. albicans* (MIC values of 0.25 to 2 µg/mL) [38].

Other groups of marine sponges produce a specialized type of chemical substance known as manzamine alkaloids, which have several important biological activities. For example, zamamidine D (**C9**), derived from the marine sponge *Amphimedon* sp., showed antimicrobial activity against various pathogens, including *C. albicans*, with a MIC of 162 µg/mL [56]. In addition, ceratinadins A (**C33**) and B (**C34**), derived from the Okinawan sponge *Pseudoceratina* sp., were active against *C. albicans* at concentrations of 2 and 4 µg/mL, respectively [42].

#### 2.1.3. Peptides

Most peptide compounds were isolated from the sponge *Theonella* sp., including theonellamide, cyclolithistide, theonegramide, and microsclerodermin. Amongst these peptides, theonellamide G (**C16**) is a new bicyclic glycopeptide from *Theonella swinhoei* that showed potential antifungal activity (MIC of 4.49 and 2.0 μM) against wild and drug-resistant strains of *C. albicans*, as well as high toxicity (6.0 μM) to the HCT-16 human colon adenocarcinoma cell line [35]. Similarly, theonellamide F (**C54**) exhibited antifungal activity against unspecific *Candida* spp. with MIC values of 3–12 µg/mL, and toxicity to leukemia cells (L1210 and P388) at an IC50 of 3.2 and 2.7 µg/mL [52]. Cyclolithistide A (**C50**), a cyclodepsipeptide, had significant antifungal activity against a reference strain of *C. albicans* (ATCC 24433), at a concentration of 202 µg/mL [49]. Additionally, theonegramide (**C53**) and microsclerodermin C (**C49**) also inhibited the growth of *C. albicans*, as reported by Bewley and Faulkner (1994) and Schmidt and Faulkner (1998), respectively [48,51]. Furthermore, there are peptides extracted from other sponges that have been reported to have antifungal activity. Discobahamin A (**C56**) and B (**C57**) (bioactive cyclic peptides) were isolated from the alcoholic extract of deep-water marine sponge *Discodermia* sp., but their antifungal activity against *C. albicans* was considered weak compared with other marine peptides [57].

#### 2.1.4. Steroids

Studies have revealed the antifungal activity of sulfated marine steroids, specifically 29-demethylgeodisterol-3-O-sulfite (**C58**) and geodisterol-3-O-sulfite (**C59**) extracted from the bioassay-guided fractionation of the extract of *Topsentia* sp., on *C. albicans* strains resistant to fluconazole. A pair of unusual antifungal molecules were reported from the marine sponge *Hippospongia lachne* with superior anticandidal activity [30]. Two sterols, 9α,11α-epoxycholest-7-ene-3β,5α,6α,19-tetrol 6-acetate (**C40**) and agosterol A (**C41**), which were isolated from the marine sponge *Dysidea arenaria*, have also shown activity against resistant *Candida* strains. These steroids showed strong inhibitory activity against efflux-mediated fluconazole-resistant strains of *C. albicans*. They directly target both MDR1 and CDR1 to reduce fluconazole resistance [45]. Another bioactive compound, 3,5-dibromo2-(3,5-dibromo-2-methoxyphenoxy) phenol (**C39**), isolated from the marine sponge *Dysidea herbacea*, exhibited significant anti-candidal activity (MIC of 7.8 µg/mL) by binding to the ergosterol of *C. albicans* and disrupting its membrane permeability. This compound also induced the leakage of potassium ions from *Candida* cells. Moreover, it displayed in vitro activity against *C. tropicalis* and *C. glabrata*, with MIC values of 7.8 µg/mL and 15.62 µg/mL, respectively [44].

#### 2.1.5. Terpenoids

Most bioactive terpenoids known are the group of phorbasins isolated from *Phorbas* sponges, including diterpenes, tetraterpenes, and sesterterpenes [58]. To date, over 11 phorbasins have been isolated and characterized with various biological activities, but only a few have exhibited antifungal activity. Two potent antifungal agents are phorboxazoles A (**C51**) and B (**C52**), which are cytostatic macrolides. These compounds have shown potent in vitro antifungal activity against *C. albicans* and *Saccharomyces carlsbergensis* at a concentration of 1 µg/disk [50]. Another example of a phorbasin with antifungal activity is phorbasin H (**C17**), which has been shown to prevent the yeast-to-hypha transition of *C. albicans* [36].

In addition to phorbasins, other terpenoids from marine sources have been identified as potential antifungal agents, such as puupehenone (**C60**) extracted from *Stronglyophora hartmani*, a deep-water marine sponge. Promisingly, puupehenone (**C60**) caused a disturbance in the fungal cell wall integrity pathway, along with Hsp90 function, and enhanced the antifungal activity of echinocandins against drug-resistant *C. albicans* and *C. glabrata* [59].

#### 2.1.6. Other Chemical Compounds

Finally, other chemical compounds from sponges showed significant antifungal activity against *C. albicans*, such as pseudoceratins A (**C36**) and B (**C37**) (bicyclic bromotyrosins) isolated from *Pseudoceratina purpurea*; pseudoceroxime A-C (**C3–C5**), and pseudocerolide D and E **(C6**, **C7**) (new bromotyrosine derivatives) extracted from *Pseudoceratina* sp. [60,61]; non-brominated racemic pyrrole derivatives from *Agelas nakamurai* [32]; and halichondramide (**C55**) from *Chondrosia corticate* [62].

### 2.2. Algae

Currently, there are over a million species of algae already known on Earth. Besides maintaining the CO_2_ levels and preventing global warming, algae are important sources of metabolites with nutritional and health benefits. Many of these metabolites present antimicrobial action and are of great interest to the pharmaceutical industry. In addition, the metabolic plasticity of algae facilitates culture development and consequently the production of pharmacological substances at a large scale. It is known that among the red, green, and brown algae, the red algae provide the greatest number of bioactive substances, such as polysaccharides (like alginate and agar), lipids, polyphenols, steroids, glycosides, flavonoids, tannins, alkaloids, and triterpenoids [29].

Indeed, most compounds with anti-candidal activity were extracted from red algae (Table 2), including Q-griffithsin (Q-GRFT) (61), a lectin derived from *Griffithsia* sp. alga, which exhibited a broad of spectrum antifungal activity against *C. albicans*, *C. glabrata*, *C. parapsilosis*, *C. krusei*, and *C. auris* [63]. In a murine model of vaginal candidiasis, Q-GRFT treatment reduced the fungal burden and enhanced the clearance of the infection without affecting immune cell phenotypes [64]. Similarly, callophycin A (**C62**), a natural product of red alga *Callophycus oppositifolius*, was found to suppress *C. albicans* growth and decrease fungal burden in vaginal candidiasis in animal models, with significant reductions in proinflammatory markers [65]. On the other hand, 10-hydroxykahukuene B (**C63**), 9-deoxyelatol (**C64**), isodactyloxene A (**C65**), and laurenmariallene (**C66**) from the species of red alga *Laurencia mariannensis* did not exhibit good anti-candidal activity [66].

Compounds extracted from green algae have also been explored for the identification of new antifungal agents. For example, the chemical extraction of the green alga *Caulerpa racemosa* resulted in two rare para-xylene derivatives caulerprenylols A (**C92**) and B (**C93**). In vitro assay revealed that caulerprenylol B (**C93**) had a broad spectrum of antifungal activity against *C. glabrata*, *Trichophyton rubrum*, and *Cryptococcus neoformans* [72].

### 2.3. Sea Cucumber

Sea cucumbers have great medicinal value in China and other Asian countries, where they have been used as tonic food for thousands of years. Currently, it is known that these animals can produce important natural products with potential antifungal action (Table 3). Among these species, triterpene glycosides are particularly noteworthy. 

A new sulfated triterpene glycoside, named coloquadranoside A (**C94**), was obtained from the sea cucumber *Colochirus quadrangularis.* This compound was effective against *C. albicans*, *C. tropicalis*, and *C. parapsilosis*, with MIC ranges of 4–25, 8–30, and 4–32 μg/mL, respectively. Interestingly, it was also found to be cytotoxic for tumor cell lines and had immunomodulatory activity [79]. Non-sulfated triterpene glycosides have also been investigated, including 10 new saponins called coustesides A–J (**C99**–**C108**) extracted from *Bohadschia cousteaui*. These compounds had antifungal action against *C. albicans*, with their zone of inhibition ranging from 10.7 ± 0.05 to 18.0 ± 0.01 [80]. Likewise, other new triterpene glycosides were extracted from *Stichopus variegates*: variegatusides C–F (**C109–C112**), variegatusides A (**C113**), B (**C114**), and holothurin B (**C133**). Amongst these compounds, variegatuside D (110) was the most effective against *C. albicans*, *C. pseudotropicalis,* and *C. parapsilosis*, with a MIC of 3.40 μg/mL [81].

Furthermore, many other triterpene glycosides with activity against *Candida* species have been identified, including arguside F (**C123**), impatienside B (**C124**), and pervicoside D (**C125**) from sea cucumber *Holothuria axiloga*; marmoratoside A (**C126**) and B (**C127**), impatienside A (**C130**), and bivittoside D (**C131**) from *Bohadschia marmorata* [83,84]; and holothurin B (**C133**) from sea cucumber *Actinopyga lecanora* [86].

### 2.4. Bacteria

Marine bacteria live in an extremely complex environment with huge diversity. The ocean column consists of approximately 10^6^ bacterial cells per milliliter of water [87]. Due to genomic adaptability to complex environments, they can exert multiple functions and produce several biologically active molecules [88]. Thereby, marine bacteria can provide sustainably active pharmacological ingredients without harming biodiversity. For these reasons, marine microbes have been recognized as a source of bioactive compounds, gaining great attention among pharmaceutical researchers.

#### 2.4.1. Actinomycetes

Actinomycetes are Gram-positive filamentous bacteria that are known for their ability to produce a wide range of bioactive compounds, including antifungal metabolites [89]. These bacteria are commonly found in soil, but they can also colonize other niches such as water, plants, and animals. The genera that produce the most commercially important biomolecules are *Streptomyces*, *Nocardia, Saccharopolyspora*, *Amycolatopsis*, *Micromonospora,* and *Actinoplanes* [90,91]. The detailed structure of some important natural products from marine Actinomycetes is presented in Figure 4 and Table 4.

Among the aforementioned genera, *Streptomyces* has gained more attention. The ability of *Streptomyces* to produce antifungal metabolites is associated with their complex genome, which contains numerous biosynthetic gene clusters that encode the production of a variety of secondary metabolites. The number of secondary metabolites has continuously increased in response to the emergence of tools and bioinformatic resources and the enhancement of deep-sea exploration technology. However, information regarding biosynthetic gene clusters still needs further investigation, such as the use of next-generation sequencing methods to obtain the genetic data of the target organisms [117]. The antifungal compounds from *Streptomyces* are mainly polyenes, macrolides, and peptides, which have potent activity against a broad spectrum of fungal pathogens [118,119,120]. Several examples are cited below.

*Streptomyces antibioticus* OUCT16-23 strain isolated from a deep-sea sediment sample produces macrolides that displayed antifungal activity against *C. albicans* [92]. *Streptomyces* sp. ZZ446 from coastal soil produces different compounds, namely streptopyrazinones A-D (**C157**–**C160**), which exhibited activity against *C. albicans* and methicillin-resistant *Staphylococcus aureus* [100,121]. Caniferolides A-D (**C148**–**C151**) from marine-derived *Streptomyces caniferus* CA-271066 showed antifungal activity against *C. albicans*, with MIC values ranging from 0.5 to 2.0 µg/mL, which were comparably lesser than the MIC of amphotericin B (2–4 µg/mL). Despite its antifungal activity, caniferolides had a high antiproliferative activity on tumor cell lines [97]. *Streptomyces xinghaiensis* SCSIO S15077 produce tunicamycin derivatives with antifungal activity against both fluconazole-resistant and sensitive *C. albicans* isolates, with emphasis on tunicamycin **C3** (**C144**), which showed MIC values of 4 and 2 μg/mL. A bioassay-guided fraction from *Streptomyces* sp. YG7 yielded two new epimers of cycloheximide with moderate activity against *C. albicans* (a MIC value of 62.5) [98,122].

Many other antifungal compounds produced by *Streptomyces* spp. have been isolated, including nitricquinomycins A–C (**C154–C156**) [99]; rocheicoside A (**C171**) [103]; 28-N-methylikaguramycin (**C176**), isoikarugamycin (**C177**), and ikarugamycin (**C178**) [106]; caboxamycin (**C190**) [115]; and piperazimycin B (**C191**) [116]. Besides *Streptomyces*, other bacteria have also been explored. The bioactive fraction of *Actinoalloateichus* sp. exhibited a broad spectrum of anti-candidal activity. Further structural characterization identified caerulomycin A (**C179**) as an active metabolite, with MIC values of 0.78–1.56 μg/mL against fluconazole-resistant *C. albicans*, 0.39–0.78 μg/mL against *C. glabrata*, and 0.78–1.56 μg/mL against *C. krusei* [107]. An Arctic-sediment-derived actinomycete, *Nocardia dassonvillei*, produces extracellular substances rich in the secondary metabolite N-(2-hydroxyphenyl)-2-phenazinamine (**C182**), which showed antifungal activity against *C. albicans*, with a MIC value of 64 µg/mL [110].

#### 2.4.2. Other Bacteria

High-throughput screening approaches have been routinely used to explore the secondary metabolites from many other marine bacteria with biological activity, as presented in Table 5. Examples include cycloprodigiosin (**C192**) (*Pseudoalteromonas rubra*), bulbimidazoles A−C (**C193**–**C195**) (*Microbulbifer* sp. DC3-6), and indolepyrazines A (**C196**) and B (**C197**) (*Acinetobacter* sp. ZZ1275), which were able to modify the susceptibility of *C. albicans* to antifungal drugs [123,124,125]. Recently, janthinopolyenemycins A (**C198**) and B (**C199**) (*Janthinobacterium* spp. ZZ145 and ZZ148) exhibited strong antifungal activity against *C. albicans* by expressing low MIC values [126]. The fermented broth of marine bacteria *Bacillus licheniformis* 09IDYM23 presented two anti-candidal glycolipids, ieodoglucomide C (**C200**) and ieodoglycolipid (**C201**), that also had activity against *C. albicans* [127]. Forazoline A (**C202**) and B (**C203**), derived from marine-invertebrate-associated bacteria, reduced the fungal burden level in mice infected with *C. albicans* [128]. Finally, the ethyl acetate extract of *Bacillus subtilis* yielded seven compounds, and all of them exhibited reasonable anti-candidal activity against *C. albicans* [129].

### 2.5. Fungi

The natural products produced by marine fungi can be classified into several groups, including alkaloids, polyketides, terpenoids, peptides, and phenolics, among others. Some of the most interesting natural products from marine fungi include cytotoxic compounds, with potential anticancer activity; immunomodulatory compounds, with potential applications in autoimmune diseases; and antimicrobial compounds, with potential applications in combating drug-resistant pathogens. The discovery of natural products from marine fungi is a rapidly growing field of research, as scientists continue to explore the vast and largely unexplored marine environment. The potential of these natural products to serve as lead compounds for drug discovery has generated significant interest, with several marine-derived compounds already in clinical trials. In addition, the sustainable production of natural products from marine fungi has the potential to provide a renewable source of bioactive compounds with minimal environmental impact. The detailed structures of some important natural products from marine fungi are presented in Figure 5 and Table 6.

#### 2.5.1. *Penicillium* spp.

*Penicillium* is a genus of fungi that includes several species known for their ability to produce a wide range of bioactive compounds, and it is the source of the first antibiotic, penicillin [157,158,159]. In particular, marine *Penicillium* species have gained increasing attention in recent years due to their unique properties to produce novel bioactive compounds [160]. They are found in various marine habitats, including sediments, mangroves, coral reefs, and seawater. A considerable number of studies have provided evidence of the bioactivity of compounds from *Penicillium* fungi; however, only few studies highlighted the importance of antifungal therapy. Here, some of the compounds with antifungal properties are reported. For example, pyrrospirones C-I (**C249–C255**) (*Penicillium* sp. ZZ380) are an uncommon class of alkaloids that inhibited the growth of *C. albicans* [147]. Melearoride A (**C260**) and B (**C266**) from *Penicillium meleagrinum* var. *viridiflavum* had activity against *C. albicans* and synergistic interaction with fluconazole against azole-resistant *C. albicans* [150]. Similar research was conducted by Kaleem et al. (2020), which resulted in the identification of 16 compounds, including andrastones B (**C277**) and C (**C278**), that had greater anti-candidal action [161].

#### 2.5.2. Endophytic Fungi

Amongst endophytic fungi, *Cladosporium* sp. was found to have two isolated compounds with biological activity: sporiolides A (**C274**) exhibited strong activity against *C. albicans*, with a concentration of 16.7 µg/mL, and sporiolides B (**C275**) showed moderate cytotoxicity against murine lymphoma L1210 cells [155]. Endophytic *Aspergillus niger* EN-13 produced nigerasperone C (**C256**), with moderate activity against *C. albicans* [148]. Furthermore, biologically active molecules didymellamide A (**C272**) and B-D (**C279–C281**) (*Stagonosporopsis cucurbitacearum*), as well as pleosporallin D (**C267**) and E (**C268**) (*Pleosporales* sp.), were obtained from different endophytic marine fungi with activity against *C. albicans* [151,153]. *Aspergillus* sp. is associated with sponge-produced tetrahydrofuran derivative known as aspericacid B (**C282**) but it exhibited no activity towards *Candida*. An another study reported that terretrione C (**C237**) from tunicate-derived fungus, *Penicillium* sp. CYE-87 was active against C. albicans with the MIC of 32 µg/mL [143]. Sponge-derived endophytic fungus *Fusarium* sp. LY019 yielded two alkaloids, fusaripyridines A (**C283**) and B (**C284**), that were identified as inhibitors of *C. albicans* growth, but they were not active against certain bacteria and HeLa cells [162]. Other promising antifungal compounds against *C. albicans* are the new thiodiketopiperazine (**C285**), epipolythiodiketopiperazine (**C286**), and trichothecene (**C287**) derivatives from *Aspergillus terreus* and *Trichoderma* cf. *brevicompactum*, respectively [140,141].

#### 2.5.3. Other Fungi

Recently, several new metabolites such as talaromydien A (**C288**) and talaroisocoumarin A (**C231**) were isolated from *Talaromyces* sp. ZZ1616. Talaroisocoumarin A (**C231**) expressed superior activity against *C. albicans* (26 µg/mL) and some bacterial species [139]. Furthermore, bioactive compounds of *Aspergillus fumigatus* were effective against *C. albicans* with a MIC of >100 µM [163]. Similar observations were made by Ding et al. and Huang et al. (2018), who proved the efficacy of secondary metabolites against *C. albicans* obtained from *Aspergillus versicolor* and *Cladosporium* sp. SCSIO z0025, respectively [144,145]. Ditalaromylectones A (**C227**) and B (**C289**), along with seven known compounds, were isolated from *Talaromyces mangshanicus* BTBU20211089, but only ditalaromylectone A (**C227**) was active against *C. albicans* [136]. *Ent*-epiheveadride (**C229**), a new nonadride enantiomer isolated from the marine fungus *Aspergillus chevalieri* PSU-AMF79, had moderate inhibitory activity against *C. albicans* with a MIC value of 200 µg/mL [137].

### 2.6. Miscellaneous

The marine sources of antifungal compounds are not limited to sponges, algae, sea cucumbers, bacteria, and fungi, but there are some other important sources from the marine environment, such as corals, mollusks, coelenterates, and bryozoans. For example, iseolide A (**C241**) (isolated from coral-derived actinomycete *Streptomyces* sp.) [94] and nocarimidazoles C (**C290**) and D (**C291**) (isolated from coral-derived actinomycete *Kocuria* sp.) [164,165] demonstrated activity against *C. albicans*. Polyketides are unique structures isolated from dinoflagellate *Amphidinium carterae*, with proven antifungal activity [166]. Similarly, *Didemnum* sp. collected from the Red Sea produce didemnaketals F (**C292**) and G (**C293**), which were able to control the growth of *C. albicans* at a concentration of 100 µg/disc with a zone of inhibition of 16–24 mm [167]. Apart from this, several studies reported some compounds with potential antifungal action from marine snail *Cenchritis muricatus* [168], sea squirt *Ciona intestinalis* [169], tunicate *Halocynthia aurantium* [170], ascidian *Clavelina oblonga* [171], and tunicate *Eudistoma* sp. [172].

## 3. Discussion and Future Perspectives

In this review, among the data collected from different studies focused on marine organisms, sponges, algae, bacteria, and fungi seem to be the major contributors of bioactive compounds with anti-candidal activity. Promisingly, marine sponges stand out due to having a large number of biologically important molecules [24]. In particular, sponges have been recognized as prolific producers of alkaloids, terpenoids, peptides, polyketides, and sterols, among others [31,32,33,36,37,38,42,44,48,49,50,51,55,56]. All these classes of molecules exhibit significant biological activities against *Candida* species [24,25,26,27,28,43,47,54].

Although natural products from sponges and other marine organisms are well known for their antifungal activities [173], many concerns still remain over their other features, including structural complexity, supply and availability, standardization and quality control, possible drug–drug interactions, side effects, toxicity, and lack of clinical evidence [11]. Among them, supply chain management is the primary concern because extracting a single molecule from a complex mixture is a long and thorough process.

To overcome the supply chain management, great attention has been given to in situ cultivation and aquaculture [11]. By adopting these methods, researchers can stimulate appropriate culture conditions without disrupting biodiversity, providing access to diverse sources of raw materials for a constant supply. In addition, both methods contribute to the standardized production of raw materials, facilitating the processes involved in the research and development of marine products.

An understanding of the complex structure of bioactive compounds can also provide insights into the production of similar kinds of chemical compounds. Chemical synthesis through which natural products are reproduced from different sources offers the option to obtain a product with comparably lesser cost than a product originating from its source [12,16,17]. 

We expect that this review can prompt researchers to establish biobanks and sample repositories of marine products with antifungal activities, promoting international collaborations and advances for the future application of marine natural products on *Candida* infections.

## 4. Conclusions

The need for new anti-candidal compounds has increased due to the emergence of various drug-resistant isolates; meanwhile, knowledge of different ecosystems can provide insights into drug discovery. The marine environment is being recognized as the treasure trove of novel chemical cues, whose potency, detailed structures, and functional properties still need to be explored as they are in terrestrial sources. 

In summary, we found that sponges, algae, and microorganisms have been the major marine sources employed to extract metabolites with potential antifungal action, although many other organisms can also provide important sources of antifungal activity. According to the studies reported, a wide number of natural compounds from the marine environment were found to be effective against clinical and reference strains of *C. albicans* and non-*albicans* species, including *C. auris*, a multi-drug-resistant species. Several compounds showed stronger antifungal activity than conventional antifungal drugs, such as fluconazole and amphotericin B. Interestingly, some of these compounds had synergistic interaction with antifungal drugs and altered the resistance mechanisms, making the *Candida* cells more susceptible to fluconazole and echinocandins. In addition to antifungal activity, certain compounds showed activity against bacteria and immunomodulatory effects, which can potentialize its effects in the treatment of candidiasis since this infection can be associated with the presence of bacteria and immunodeficiency.

Although many antifungal compounds had already been isolated from marine organisms, most studies are limited to verifying their antifungal activity in in vitro models. To translate these compounds into clinical applications, there is still a long way to go, with the development of in vivo studies, toxicity assays, and investigations of action mechanisms. Moreover, some marine organisms are protected by international law in specific regions of the world, and it is uncertain whether there are enough raw materials to ensure a steady supply of natural products. Thus, new approaches are needed to address the issues related to the sustainable production and marketing of natural products using contemporary technologies to preserve maritime ecosystems.

## Figures and Tables

**Figure 1 jof-09-00800-f001:**
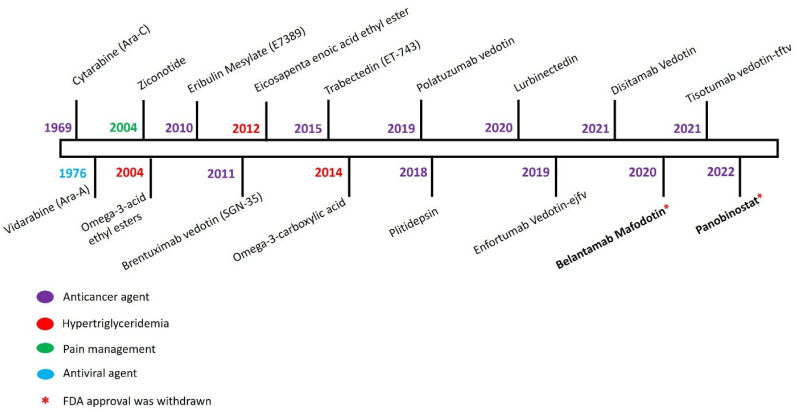
List of FDA-approved natural products from marine resources and their year of approval. (Source: https://www.marinepharmacology.org/, accessed on 10 July 2023).

**Figure 2 jof-09-00800-f002:**
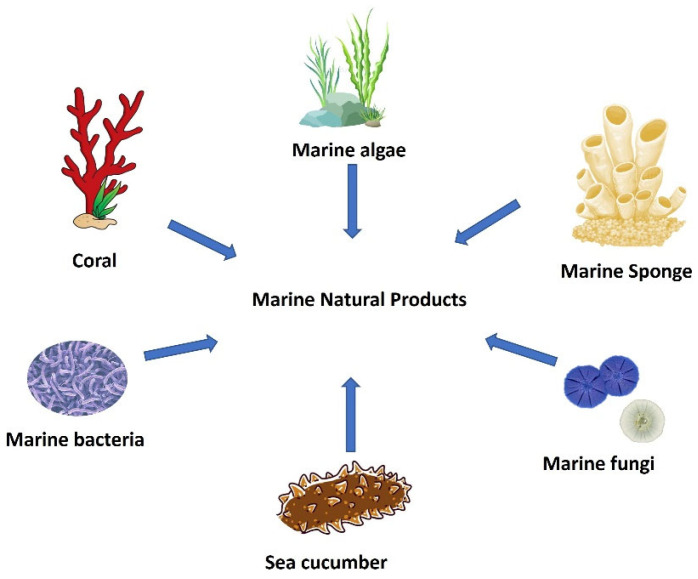
Marine organisms for the isolation of different marine natural products.

**Figure 3 jof-09-00800-f003:**
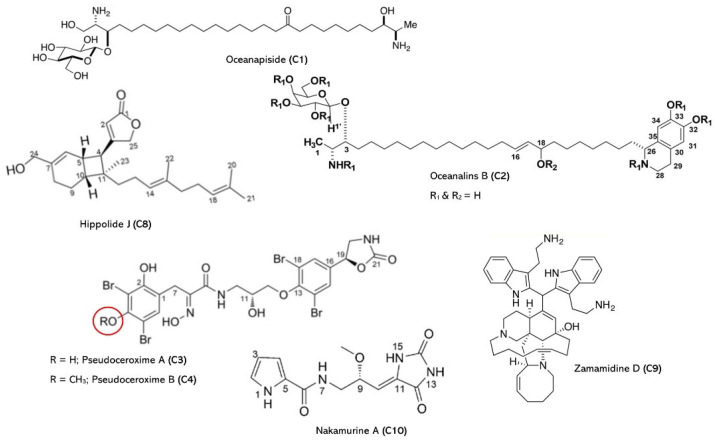
Marine natural products from marine sponges. Red circle is an indicative of R group.

**Figure 4 jof-09-00800-f004:**
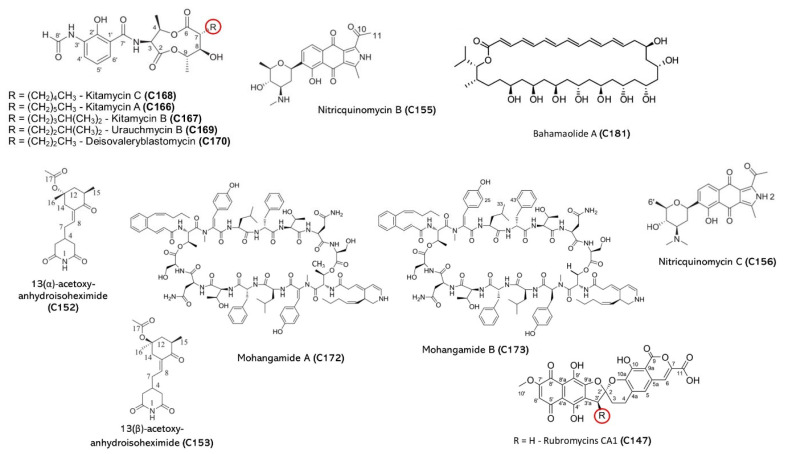
Marine natural products from marine actinomycetes. Red circle is an indicative of R group.

**Figure 5 jof-09-00800-f005:**
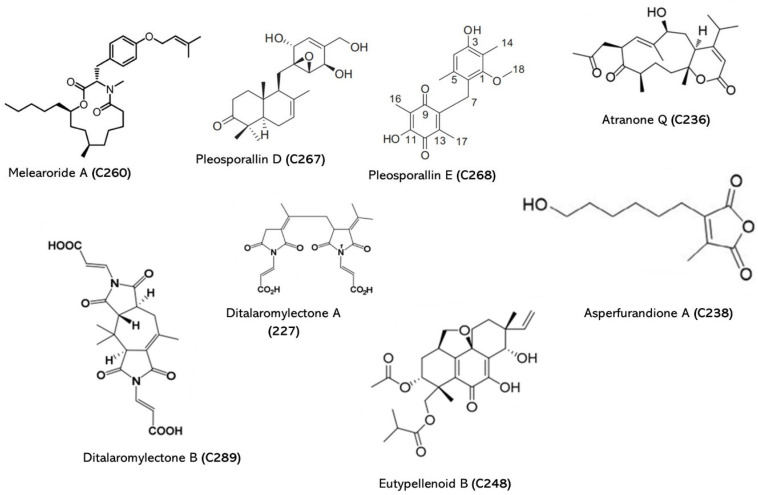
Marine natural products from marine fungi.

**Table 1 jof-09-00800-t001:** Natural products isolated from the samples of marine sponge and their activity against different *Candida* spp.

	Compound	Source	ZOI (mm)	MIC (µg/mL) and Activities	Target Organism	Reference
**C1**	Oceanapiside	*Oceanapia phillipensis*		10	Cg	Dalisay et al., 2021 [27]
**C2**	Oceanalin B	*Oceanapia* sp.		25	Cg	Makarieva et al., 2021 [28]
**C3**	Pseudoceroxime A	*Pseudoceratina* sp.		11.9	Ca	Chen et al., 2020 [29]
**C4**	Pseudoceroxime B			13		
**C5**	Pseudoceroxime C			19.80		
**C6**	Pseudoceroxime D			>20		
**C7**	Pseudoceroxime E			>20		
**C8**	Hippolide J	*Hippospongia lachne*		0.125–4	Ca, Cp, and Cg	Jiao et al., 2017 [30]
**C9**	Zamamidine D	*Amphimedon* sp.		16	Ca	Kubota et al., 2017 [31]
**C10**	Nakamurine A	*Agelas nakamurai*			Ca	Chu et al., 2017 [32]
**C11**	Nakamurine B			60	Ca	Chu et al., 2017 [32]
**C12–C14**	(Z)-5-(4-hydroxybenzylidene)-imidazolidine-2,4-dione, hemimycalins A and B	*Hemimycale arabica*	22, 14, and 20		Ca	Youssef et al., 2015 [33]
**C15**	Crambescin	*Pseudaxinella reticulate*		11–39; 6.1–17; 11–34	Ca, Cg, and Ck	Jamison and Molinski, 2015 [34]
**C16**	Theonellamide G	*Theonella swinhoei*		4.49 and 2.0	Ca	Youssef et al., 2014 [35]
**C17**	Phorbasin H			250 Targeting of yeast-to-hypha transition	Ca	Lee et al., 2013 [36]
**C18–C19**	Nagelamide U and W	*Agelas* sp.		4	Ca	Tanaka et al., 2013a [37]
**C20–C22**	Nagelamide X-Z	*Agelas* sp.		0.25 to 2	Ca	Tanaka et al., 2013b [38]
**C23**	Aurantoside K	*Melophlus*		31.25 and 1.95	Ca	Kumar et al., 2012 [39]
**C24**	Agelasine O	*Agelas* sp.		>32	Ca	Kubota et al., 2012 [31]
**C25**	Agelasine P			>32		
**C26**	Agelasine Q			16		
**C27**	Agelasine R			16		
**C28**	Agelasine S			>32		
**C29**	Agelasine T			>32		
**C30**	Agelasine U			>32		
**C31**	Woodylide A	*Plakortis simplex*		32	Ca	Yu et al., 2012 [40]
**C32**	Aurantoside J	*Theonella swinhoei*		>16	Ca, Cp, Cg, and Ct	Angawi et al., 2011 [41]
**C33**	Ceratinadin A	*Pseudoceratina* sp.		2	Ca	Kon et al., 2010 [42]
**C34**	Ceratinadin B			4		
**C35**	Ceratinadin C			>32		
**C36–C37**	Pseudoceratin A and B	*Pseudoceratina purpurea*	8 and 6.55	-	Ca	Jang et al., 2007 [39]
**C38**	Oceanalin A	*Oceanapia* sp.		30	Cg	Makarieva et al., 2005 [43]
**C39**	3,5-dibromo2-(3,5-dibromo-2-methoxyphenoxy)	*Dysidea herbacea*		7.8, 7.8; 15.62	Ca, Ct, and Cg	Sionov et al., 2005 [44]
**C40–C41**	9α,11α-epoxycholest-7-ene-3β,5α,6α,19-tetrol 6-acetate and agosterol A	*Dysidea arenaria*		Targeting of MDR1 and CDR1	Ca	Jacob et al., 2003 [45]
**C42–C43**	Bengamide Bengazole	*Pachastrissa* sp.		0.8 to 1.5	Ca	Fernández et al., 1999 [46]
**C44**	Aurantoside A	*Siliquariaspongia japonica*		1.25	Ca	Sata et al., 1999 [47]
**C45**	Aurantoside B			0.63		
**C46**	Aurantoside D			9.5		
**C47**	Aurantoside E			0.16		
**C48**	Aurantoside F			Inactive		
**C49**	Microsclerodermin C	*Theonella* sp.		5	Ca	Schmidt, E. W., and Faulkner, 1998 [48]
**C50**	Cyclolithistide A	*Theonella swinhoei*		20	Ca	Clark et al., 1998 [49]
**C51–C52**	Phorboxazoles A and B	*Phorbas* sp.	12		Ca	Searle et al., 1995 [50]
**C53**	Theonegramide	*Theonella swinhoei*	10		Ca	Bewley, C. A., and Faulkner, 1994 [51]
**C54**	Theonellamide F	*Theonella* sp.		3–12	unspecific *Candida* spp.	Matsunaga et al., 1989 [52]
**C55**	Halichondramide	*Halichondria* sp.		0.2	Ca	Kernan et al., 1987 [53]

Ca: *Candida albicans*; Ct: *Candida tropicalis*; Ck: *Candida krusei*; Cp: *Candida parapsilosis*; Cg: *Candida glabrata;* ZOI—zone of inhibition; MIC—minimum inhibitory concentration.

**Table 2 jof-09-00800-t002:** Natural products isolated from the samples of marine algae and their activity against different *Candida* spp.

	Compound	Source	ZOI (mm)	MIC (µg/mL) and Activities	Target Organism	Reference
	Crude extract	*Champia parvula*	13.8 ± 0.08 and 16.7 ± 0.15		Ca, Ct	Ganesan 2019 [67]
**C67**	(9Z,12Z,15Z,18Z,21Z)-ethyl tetracosa-9,12,15,18,21- pentaenoate	*Laurencia okamurai*		4	Cg	Feng et al., 2015 [68]
**C68**	Mahorone	*Asparagopsis taxiformis*		>32	Ca	Greff et al., 2014 [69]
**C69**	5-bromomahorone			>32		
**C70**	Laurepoxyene	*Laurencia okamurai*		2	Cg	Yu et al., 2014 [70]
**C71**	3b-Hydroperoxyaplysin			4		
**C72**	3a-Hydroperoxy-3-epiaplysin			>64		
**C73**	8,10-Dibromoisoaplysin			>64		
**C74**	(5S)-5-Acetoxy-b-bisabolene			64		
**C75**	10-Bromoisoaplysin			32		
**C76**	Laurokamurene C			1		
**C77**	Laurokamurene A			64		
**C78**	Phlorotannin	*Cystoseira nodicaulis Cystoseira usneoides and Fucus spiralis*		15.6 and 31.3; 31.3 and >62.5	Ca; Ck	Lopes et al., 2013 [71]
**C79**	Caulerprenylol B	*Caulerpa racemosa*		4	Cg	Liu et al., 2013 [72]
**C80**	Bromophycolide U			>15	Ca	Lin et al., 2010 [73]
**C81**	Isolauraldehyde	*Laurencia obtusa*		70	Ca	Alarif et al., 2012 [74]
**C82**	12-hydroxy isolaurene			2000		
**C83**	8,11-dihydro-12-hydroxy isolaurene			120		
**C84**	Symphyocladin G	*Symphyocladia latiuscula*		10	Ca	Xu et al., 2012 [75]
**C85**	Bromophycolide R	*Callophycus serratus*		>15	Ca	Lin et al., 2010 [73]
**C86**	Bromophycolide S			>15		
**C87**	Bromophycolide T			>15		
**C88**	2,20,3,30-tetrabromo-4,40,5,50-tetrahydroxydiphenylmethane	*Odonthalia corymbifera*		1.56	Ca	Oh et al., 2008 [76]
**C89–C90**	Capisterones A and B	*Penicillus capitatus*		CDR1 efflux pump activity	Ca	Li et al., 2006 [77]
**C91**	Acetoxyfimbrolide	*Delisea pulchra*	17		Ca	Ankisetty et al., 2004 [78]

Ca: *Candida albicans*; Ct: *Candida tropicalis*; Ck: *Candida krusei*; Cp: *Candida parapsilosis*; Cg: *Candida glabrata;* ZOI—zone of inhibition; MIC—minimum inhibitory concentration.

**Table 3 jof-09-00800-t003:** Natural products isolated from the samples of sea cucumber and their activity against different *Candida* spp.

	Compound	Group	Source	ZOI (mm)	MIC (µg/mL)	Target Organism	Reference
**C94** **C95** **C96** **C97** **C98**	Coloquadranoside A Philinopside A Philinopside B Philinopside E Pentactaside B	Triterpene glycosides	*Colochirus quadrangularis*		4, 8, 4 20, 30, 32 4, 8, 4 4, 8, 4 25	Ca, Ct, and Cp	Yang et al., 2021 [79]
**C99–C108**	Cousteside A-J	Non-sulfated triterpene glycosides	*Bohadschia cousteaui*	10.7 ± 0.05 to 18.0 ± 0.01		Ca	Elbandy et al., 2014 [80]
**C109** **C110** **C111** **C112** **C113** **C114**	Variegatuside C Variegatuside D Variegatuside E Variegatuside F Variegatuside A Variegatuside B	Triterpene glycosides	*Stichopus variegates* Semper		12.5; 25; 12.5 3.4; 3.4; 13.6 25; 12.5; 12.5 25; 12.5; 12.5 25; 12.5; 12.5 100; 25; >125	Ca, Cp, and Ct	Wang et al., 2014 [81]
**C115**	26-Nor-25-oxo-holotoxin A1	Triterpene glycosides	*Apostichopus japonicus* Selenka		>45.91	Ca and Ct	Wang et al., 2012 [82]
**C116**	Holotoxin D				6.64, 13.29		
**C117**	Holotoxin E				13.45, 13.45		
**C118**	Holotoxin F				5.58, 5.68		
**C119**	Holotoxin G				5.81, 5.81		
**C120**	Holotoxin A1				11.49, 5.68		
**C121**	Holotoxin B				11.36, 5.68		
**C122**	Cladoloside B				3.28, 1.64		
**C123**	Arguside F	Triterpene glycosides	*Holothuria (Microthele) axiloga*		64, 16, 16	Ca, Ct, and Ck	Yuan et al., 2009a [83]
**C124**	Impatienside B				4, 4, 4		
**C125**	Pervicoside D				64, 16, 16		
**C126** **C127** **C128** **C129** **C130** **C131**	Marmoratoside A Marmoratoside B 17α-hydroxy impatienside A 25-acetoxy bivittoside D Impatienside A Bivittoside D	Triterpene glycosides	*Bohadschia marmorata*		2.81; 2.81; 11.24 2.78; 2.78; 2.78 44.44; 44.44; 44.44 43.13; 10.78; 10.78 2.81; 2.81; 2.81 2.80; 2.80; 2.80	Ca, Ct, and Ck	Yuan et al., 2009b [84]
**C132**	Axilogoside A (**132**)	Triterpene glycoside	*Holothuria (Microthele) axiloga*		16	Ca	Wei-Hua et al., 2008 [85]
**C133**	Holothurin B (**133**)	Triterpene glycoside	*Actinopyga lecanora*		25, 12.5 and 6.25	Ca, Ck, and Cp	Kumar et al., 2007 [86]

Ca: *Candida albicans*; Ct: *Candida tropicalis*; Ck: *Candida krusei*; Cp: *Candida parapsilosis*; Cg: *Candida glabrata;* ZOI—zone of inhibition; MIC—minimum inhibitory concentration.

**Table 4 jof-09-00800-t004:** Natural products isolated from the samples of marine actinomycetes and their activity against different *Candida* spp.

	Compound	Group	Source	ZOI (mm)	MIC (µg/mL) and Activities	Target Organism	Reference
**C134** **C135** **C136** **C137** **C138** **C139**	Chainin Filipin IX Filipin XI Filipin XII Filipin II Filipin III	Deep-sea actinobacteria	*Streptomyces antibioticus OUCT16-2*		1.56–12.5	Ca	Bao et al., 2022 [92]
**C140**	Antimycin I	Sponge-associated	*Streptomyces* sp. *NBU3104*		8	Ca	Li et al., 2022 [93]
**C141–C143**	Iseolide A–C	Coral-derived	*Streptomyces* sp.		0.39–6.25	Ca	Zhang et al., 2020 [94]
**C144**	Tunicamycin C3	Deep sea	*Streptomyces xinghaiensis SCSIO S15077*		4–32	Ca	Zhang et al., 2020 [95]
**C145**	Maculosin	Costa soil	*Streptomyces* sp. *ZZ446*		27	Ca	Chen et al., 2020 [61]
**C146**	Maculosin-O-a-L rhamnopyranoside				26		
**C147**	Rubromycin CA1	Tunicate	*Streptomyces hyaluromycini*		6.3	Ca	Harunari et al., 2019 [96]
**C148–C151**	Caniferolide A-D		*Streptomyces caniferus CA-271066*		0.5 to 2.0	Ca	Pérez-Victoria et al., 2019 [97]
**C152–** **C153**	13(α)-Acetoxy-anhydroisoheximide and 13(β)-acetoxy-anhydroisoheximid	Deep sea	*Streptomyces* sp. *YG7*		62.5	Ca	Pan 2019 [98]
**C154–C156**	Nitricquinomycin A-C	Marine-sediment-derived	*Streptomyces* sp. *ZS-A45*		>40	Ca	Zhou et al., 2019 [99]
**C157–C160**	Streptopyrazinone A-D	Costal soil	*Streptomyces* sp. *ZZ446*		35–60	Ca	Chen et al., 2018 [100]
**C161**	N-acetyl-L-isoleucine-L-leucinamide						
**C162–C165**	Strepoxepinmycin A–D	Marine environment	*Streptomyces* sp. *XMA39*		5 to 10	Ca	Jiang et al., 2018 [101]
**C166–C168**	Kitamycin A-C		*Streptomyces antibioticus strain 200-09*		25	Ca	Wang et al., 2017 [102]
**C169**	Urauchmycin B						
**C170**	Deisovaleryblastomycin						
**C171**	Rocheicoside A	Marine-sediment-derived	*Streptomyces rochei 06CM016*	37		Ca	Aksoy et al., 2016 [103]
**C172–C173**	Mohangamide A and B	Marine actinomycete	*Streptomyces* sp.	inhibiting isocitrate lyase	IC_50_ = 4.4 and 20.5 µM	Ca	Bae et al. 2015 [104]
**C174–C175**	Reedsmycin A and F		*Streptomyces* sp. *CHQ-64*		25–50	Ca	Che et al., 2015 [105]
**C176**	28-*N*-Methylikaguramycin	Marine sediment	*Streptomyces zhaozhouensis CA-185989*		4	Ca	Lacret et al., 2014 [106]
**C177**	Isoikarugamycin				2–4		
**C178**	Ikarugamycin				4		
**C179**	Caerulomycin A	Marine actinomycetes	*Actinoalloateichus cyanogriseus*		0.39–0.78; 0.78–1.56	Ca, Cg, and Ck	Ambavane et al., 2014 [107]
**C180**	Arcticoside	Arctic Actinomycete	*Streptomyces* sp.	Inhibition of *C. albicans* Isocitrate Lyase	30.4 μM	Ca	Moon et al., 2014 [108]
**C181**	Bahamaolide A	Marine actinomycete	*Streptomyces* sp.		12.5	Ca	Kim et al. 2012 [109]
**C182**	(*N*-(2-hydroxyphenyl)-2-phenazinamine)	Arctic sediment	*Nocardia dassonvillei*		64	Ca	Gao et al., 2012 [110]
**C183**	Azalomycin F4a 2-ethylpentyl ester	Mangrove rhizosphere soil	*Streptomyces* sp. *211726*		2.34 and 12.5	Ca	Yuan et al., 2013 [111]
**C184**	Azalomycin F5a 2-ethylpentyl ester						
**C185–C186**	Antimycins A_19_ and A_20_		*Streptomyces antibioticus H74-18*		5 to 10	Ca	Xu et al., 2011 [112]
**C187**	Saadamycin	Egyptian sponge *Aplysina fistularis*	*Streptomyces* sp. *Hedaya48*		2.22 and 15	Ca	El-Gendy and EL-Bondkly, 2010 [113]
**C188**	5,7-Dimethoxy-4-p-methoxylphenylcoumarin						
**C189**	Chitinase	Sponge associate	*Streptomyces* sp. *DA11*	10.48 ± 0.45	-	Ca	Han et al., 2009 [114]
**C190**	Caboxamycin	Deep sea cold water	*Streptomyces* sp. *NTK 937*	-	117	Cg	Hohmann et al., 2009 [115]
**C191**	Piperazimycin B	Marine-derived	*Streptomyces* sp.	14	-	Ca	Shaaban et al., 2008 [116]

Ca: *Candida albicans*; Ct: *Candida tropicalis*; Ck: *Candida krusei*; Cp: *Candida parapsilosis*; Cg: *Candida glabrata;* ZOI—zone of inhibition; MIC—minimum inhibitory concentration.

**Table 5 jof-09-00800-t005:** Natural products isolated from the samples of marine bacteria and their activity against different *Candida* sp.

	Compound	Group	Source	ZOI (mm)	MIC (µg/mL)	Target Organism	Reference
**C192** **C204** **C205**	Cycloprodigiosin Prodigiosin 2-Methyl-3-hexyl prodiginine	Red marine bacterium	*Pseudoalteromonas rubra*	7.9 ± 0.07, 8.2 ± 0.09 7.9 ± 0.06		Ca	Setiyono et al., 2020 [123]
**C193–C195**	Bulbimidazole A−C		*Gammaproteobacterium Microbulbifer*		6.25–12.5	Ca	Karim et al., 2020 [124]
**C196** **C197**	Indolepyrazine A Indolepyrazine B	Marine bacteria	*Acinetobacter* sp. *ZZ1275*		12 and 14	Ca	Anjum et al., 2019 [125]
**C198** **C199**	Janthinopolyenemycin A Janthinopolyenemycin B	Marine bacteria	*Janthinobacterium* spp. *ZZ145 and ZZ148*		15.6	Ca	Anjum et al., 2018 [126]
**C200** **C201**	Ieodoglucomide C Ieodoglycolipid	Marine bacteria	*Bacillus licheniformis 09IDYM23*		0.05 and 0.03	Ca	Tareq et al., 2015 [127]
**C202** **C203**	Forazoline A Forazoline B	Invertebrate-associated bacteria			16	Ca	Wyche et al., 2014 [128]
**C204–C210**	Gageomacrolactin A-C, macrolactins A (C207), B (C208), E (C209) and W (C210)	Marine sediments	*Bacillus subtilis*		0.05–0.15	Ca	Tareq et al., 2013 [129]
**C211–218**	Quinazolinones (in total 8 analogues)	Marine bacterium	*Bacillu cereus 041381*		1.3–15.6	Ca	Xu et al., 2011 [130]
**C219**	Pedein A		*Chondromyces pediculatus*	32	1.6	Ca	Kunze et al. 2008 [131]
**C220**	2-Nitro-4-(2′-nitroethenyl)-phenol	Arctic sea ice bacterium	*Salegentibacter* sp. *T436*		20	Ca	Al-Zereini et al. 2007 [132]
**C221**	Hassallidin A	Cyanobacterium	*Hassallia* sp.		4.8	Ca	Neuhof et al., 2005 [133]
**C222** **C223**	Basiliskamide A Basiliskamide B	Tropical marine habitat	*Bacillus laterosporus*		1.0 3.1	Ca	Barsby et al., 2002 [134]
**C224–C226**	Lobocyclamide A-C	Cyanobacterial mat	*Lyngbya confervoides*	7–10 and 6–8		Ca and Cg	MacMillan et al., 2002 [135]

Ca: *Candida albicans*; Ct: *Candida tropicalis*; Ck: *Candida krusei*; Cp: *Candida parapsilosis*; Cg: *Candida glabrata;* ZOI—zone of inhibition; MIC—minimum inhibitory concentration.

**Table 6 jof-09-00800-t006:** Natural compounds isolated from the samples of marine fungi and their activity against different *Candida* spp.

	Compound	Group	*Source*	ZOI (mm)	MIC (µg/mL)	Target organism	Reference
**C227** **C228**	Ditalaromylectone A Altenusin		*Talaromyces mangshanicus BTBU20211089*		200	Ca	Zhang et al., 2022 [136]
**C229**	*ent*-Epiheveadride	Marine sediment	*Aspergillus chevalieri PSU-AMF79*		200	Ca	Ningsih et al., 2022 [137]
**C230**	(-)-Massoia lactone	Unidentified tunicate	*Trichoderma harzianum PSU-MF79*		200	Ca	Nuansri et al., 2021 [138]
**C231**	Talaroisocoumarin A	Marine-derived fungi	*Talaromyces* sp. *ZZ1616*		26	Ca	Mingzhu Ma et al., 2020 [139]
**C232**	Emethacin C	Tissue of sea hare aplysia pulmonica	*Aspergillus terreus*		32	Ca	Wu et al., 2020 [140]
**C233–C235**	Trichobreol A-C	Marine red alga	*Trichoderma* cf. *brevicompactum*		3.1- 50	Ca	Yamazaki et al. 2020 [141]
**C236**	Atranone Q	Marine fungus	*Stachybotrys chartarum*		8	Ca	Yang et al., 2019 [142]
**C237**	Terretrione C	Tunicate-derived fungus	*Penicillium* sp.	19	32	Ca	Shaala LA and Youssef DT 2015 [143]
**C238** **C239**	Asperfurandione A Asperfurandione B	Deep-sea fungi	*Aspergillus versicolor*		64	Ca	Ding et al., 2019 [144]
**C240–C247**	Cladosporiumin A-H		*Cladosporium* sp. *SCSIO z0025*		Anti-biofilm	Ca	Huang et al., 2018 [145]
**C248**	Eutypellenoid B	Arctic fungus	*Eutypella* sp.		8, 16 and 32	Ca, Cg, and Ct	Yu et al., 2018 [146]
**C249–C255**	Pyrrospirones C-I	Marine-derived fungus	*Penicillium* sp. *ZZ380.*		No activity	Ca	Song et al., 2018 [147]
**C256**	Nigerasperone C	Marine brown alga	*Aspergillus niger EN-13*	9.0	-	Ca	Zhang et al., 2007 [148]
**C257**	Penicillenol	Marine sediment	*Aspergillus restrictus DFFSCS006*		>200	Ca	Wang et al., 2017 [149]
**C258** **C259**	Penicillenol A2 Penicillenol B1	Marine sediment	*Aspergillus restrictus DFFSCS006*		Inhibit the biofilm growth and hyphae-related genes	Ca	Wang et al., 2017 [149]
**C260**	Melearoride A	Marine-derived fungus	*Penicillium meleagrinum var. viridiflavum*		>32	Ca	Okabe et al., 2016 [150]
**C261**	PF1163A				1		
**C262**	PF1163B				2		
**C263**	PF1163D				>32		
**C264**	PF1163H				16		
**C265**	PF1163F				8		
**C266**	Melearoride B				>32		
**C267** **C268**	Pleosporallin D Pleosporallin E	Marine alga *Enteromorpha clathrata*	*Pleosporales* sp.		>10 7.44	Ca	Chen et al., 2015 [151]
**C269**	Dendrodochol A	Sea cucumber	*Dendrodochium* sp.		16, 16, 16	Ca, Cp, and Cg	Xu et al., 2014 [152]
**C270**	Dendrodochol B				16, >64, >64		
**C271**	Dendrodochol C				16, 16, 8		
**C272**	Dendrodochol D				> 64 all		
**C273**	Didymellamide A	Marine-sponge-associated	*Stagonosporopsis cucurbitacearum*		3.1; 3.1	Ca and Cg	Haga et al., 2013 [153]
**C274**	Citrafungin A	Marine fungi	*Aspergillus aculeatus*		0.43	Ca	Singh 2004 [154]
**C275** **C276**	Sporiolide A Sporiolide B	Brown alga *Actinotrichia fragilis*	*Cladosporium* sp.		16.7 >33.33	Ca	Shigemori et al., 2004 [155]
**C277**	Xestodecalactone A	Marine-sponge-associated	*Penicillium* cf. *montanense*	7, 12 and 25	20, 50 and 100	Ca	Edrada et al., 2002 [156]

Ca: *Candida albicans*; Ct: *Candida tropicalis*; Ck: *Candida krusei*; Cp: *Candida parapsilosis*; Cg: *Candida glabrata;* ZOI—zone of inhibition; MIC—minimum inhibitory concentration.

## Data Availability

Data is contained within the article.

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
