# Peer review of "Anti-Candidal Marine Natural Products: A Review"

_jof, 2023, doi:10.3390/jof9080800_

Round 1
Reviewer 1 Report
SourceURL:file:///mnt/4081F70214B715EB/Laboratory/Reviewing/ANTI-CANDIDAL MARINE NATURAL PRODUCTS - A REVIEW/ANTI-CANDIDAL MARINE NATURAL PRODUCTS - A REVIEW.docx
This review paper entitled “Anti-Candidal Marine Natural Products: A Review” submitted by Arumugam and all., presents marine natural products identified as active compounds against Candida sp. pathogens, especially C. albicans. Authors listed bioactive compounds based on their origin (sponges, fungi,...) and their class (alkaloids, peptides,...) and present them under table-format, summarizing their origin and MIC. The manuscript is clear and well written. However, in its present form, the manuscript needs major modifications before acceptance for publication, especially in terms of the quality of the figures, compilation of data and discussion.
Comments and modifications:
In this review, many molecules are listed in table format and some of them are more discussed in your manuscript. It would be more easy for the reader if all the molecules are numbered in the same listing (1 to n) and not resetting the numbering for each source. I would also highly recommend to include the numbering in your text after mentioning the molecule as it would be more easy to refer to it in the table with the information. As you also present some chemical structures, please also include the respective number in your figures.
When using the term “anti-candidal” I do not think that “candidal” should be in italic.
Figure 1:
I highly suggest to redesign your timeline respecting a time scale. Your figure is informative but it will highlight the challenges and recent successes about the discovery of marine natural products as antifungal agents. As many agents have been recently discovered, you can use stretched lines to show the recent discoveries. Please also check your mention for Panobinostat (2022?). I would also recommend to use some color codes to provide more information such as organism source, clinical trials, class of compounds,...
Figure 2:
Why did you decide to show this central molecule? Is there any specific reason? Is this molecules common to all marine organisms? And what it the name of this molecule? Please also check your figure title, especially the redundant “marine” word (same comment for all your figure titles).
Figure 3 - 4:
As I mentioned above, please include a number to identify all the molecules you mention and present in your manuscript, it would be more easy for the reader (and reviewers) to look at the information in your manuscript.
Please improve the resolution - size of your figures, chemical structures are very small and difficult to read.
Figure 4:
Please check structure for compounds mohangamide A and B, what is the meaning/position of R2?
Table 1 - 6:
You collected a lot of information and data for your review article but you “just” display them without any analysis or discussion. Your work would be more original and would bring more information if you sort or organize your data in order to extract and show some trends in anti-candidal compounds from marine origin. Why did you decide to list the molecules in such order? Sorting molecules based on class of compounds, origin - genus, target organism, extract - pure compounds or any rational of your choice would definitely add value to your work. Also pay attention when you use the term “phytochemicals” (as plural, please correct) for presenting data from sponges and microorganisms,... as they do not belong to the Planta classification Kingdom.
Table 3:
Why molecules 13 and 14 are striked through?
Line 16:
You wrote “the present review attempts to discuss the significance....” but after reading your manuscript there is no attempt and even no discussion about the potential of marine natural products as anti-candidal agents. You collected a lot of information but as I mentioned above, you only display the data you found. Your work would have much more recognition and scientific values if you analyze data, is there any trends, potential - promising candidates that you could comment using your expertise. You should discuss the data you collected, are some organisms more interesting to focus on (class of molecules?) for new drugs, how these compounds specifically target Candida sp.? Is there any protein target which is more interesting or specific? What are the future challenges and directions? As you mentioned in your conclusion, some marine organisms are legally protected or endangered. What if someone discover the “perfect molecule” and this molecule is maybe produced at a very low titer, are we planning to intensively grow this organism? Is there any current research about in vitro biosynthesis in model hosts for production at industrial scale? Please bring, more discussion and analysis about the significance of marine natural products as anti-candidal agents.
Minor editing of English language required and double proof-reading.
Author Response
Reviewer 1
This review paper entitled “Anti-Candidal Marine Natural Products: A Review” submitted by Arumugam and all., presents marine natural products identified as active compounds against Candida sp. pathogens, especially C. albicans. Authors listed bioactive compounds based on their origin (sponges, fungi,...) and their class (alkaloids, peptides,...) and present them under table-format, summarizing their origin and MIC. The manuscript is clear and well written. However, in its present form, the manuscript needs major modifications before acceptance for publication, especially in terms of the quality of the figures, compilation of data and discussion.
Comments and modifications:
In this review, many molecules are listed in table format and some of them are more discussed in your manuscript. It would be more easy for the reader if all the molecules are numbered in the same listing (1 to n) and not resetting the numbering for each source. I would also highly recommend to include the numbering in your text after mentioning the molecule as it would be more easy to refer to it in the table with the information. As you also present some chemical structures, please also include the respective number in your figures.
1. When using the term “anti-candidal” I do not think that “candidal” should be in italic.
Answer: As suggested by reviewer the term ”anti-candidal” is presented appropriately in the revised manuscript.
2. Figure 1: I highly suggest to redesign your timeline respecting a time scale. Your figure is informative but it will highlight the challenges and recent successes about the discovery of marine natural products as antifungal agents. As many agents have been recently discovered, you can use stretched lines to show the recent discoveries. Please also check your mention for Panobinostat (2022?). I would also recommend to use some color codes to provide more information such as organism source, clinical trials, class of compounds,...
Answer: As suggested by the reviewer, Figure 1 was modified and presented the same in the revised manuscript. The information provided in Fig 1 is collective information on FDA-approved natural products from marine resources. To date, there is no approved anti-fungal or anti-candidal molecule from the marine environment. That’s why we could not be able to provide the information about the FDA-approved antifungals. Apart from antifungals, the approved molecules are used to treat commonly for cancer, a few for hyperlipidemia, and only one for pain management and antiviral activity. Based on their bioactivity, all the molecules are differentiated with the help of different colours. The position of “Panobinostat” was modified. Recently, FDA approval of Belantamab Mafodotin and Panobinostat was withdrawn. So, all the information is updated in the revised figure.
3. Figure 2: Why did you decide to show this central molecule? Is there any specific reason? Is this molecules common to all marine organisms? And what it the name of this molecule? Please also check your figure title, especially the redundant “marine” word (same comment for all your figure titles).
Answer: There is no common relationship of this central molecule among the sources. It is presented just for the projection purpose. Hence the central molecule was removed from the revised Figure 2 as suggested by the reviewer.
4. Figure 3 - 4: As I mentioned above, please include a number to identify all the molecules you mention and present in your manuscript, it would be more easy for the reader (and reviewers) to look at the information in your manuscript.
Answer: All the compounds discussed in this manuscript were consecutively numbered to meet requirement of the readership.
5. Please improve the resolution - size of your figures, chemical structures are very small and difficult to read.
Answer: As suggested by the reviewer chemical structures presented in this manuscript was revised and all figures are presented in 300dpi quality as specified in the journal requirement.
6. Figure 4: Please check structure for compounds mohangamide A and B, what is the meaning/position of R2?
Answer: To avoid the confusions between the structures of mohangamide A and B both are presented separately.
7. Table 1 - 6: You collected a lot of information and data for your review article but you “just” display them without any analysis or discussion. Your work would be more original and would bring more information if you sort or organize your data in order to extract and show some trends in anti-candidal compounds from marine origin. Why did you decide to list the molecules in such order? Sorting molecules based on class of compounds, origin - genus, target organism, extract - pure compounds or any rational of your choice would definitely add value to your work. Also pay attention when you use the term “phytochemicals” (as plural, please correct) for presenting data from sponges and microorganisms,... as they do not belong to the Planta classification Kingdom.
Answer: Thank you for your valuable suggestion for the improvement of this manuscript. When we attempted to prepare this manuscript, we planned to segregate the data based on the chemical class of the molecules. Unfortunately, we could not present them appropriately because of fewer compounds from the same group (i.e alkaloids, terpenoids, peptides, polyketides). Because of these reasons, we have sorted the molecules based on the origin of species (like sponge, sea cucumbers, and marine microbes). In the table, all the compounds are separately presented in the same manner based on their year of publication (reverse chronological order). We felt that this would be more convenient to present this data rather than presenting based on the class of molecules.
8. Table 3: Why molecules 13 and 14 are striked through?
Answer: It happed due to typo error. Table 3 was revised and presented appropriately.
9. Line 16: You wrote “the present review attempts to discuss the significance....” but after reading your manuscript there is no attempt and even no discussion about the potential of marine natural products as anti-candidal agents. You collected a lot of information but as I mentioned above, you only display the data you found. Your work would have much more recognition and scientific values if you analyze data, is there any trends, potential - promising candidates that you could comment using your expertise. You should discuss the data you collected, are some organisms more interesting to focus on (class of molecules?) for new drugs, how these compounds specifically target Candida ? Is there any protein target which is more interesting or specific? What are the future challenges and directions? As you mentioned in your conclusion, some marine organisms are legally protected or endangered. What if someone discover the “perfect molecule” and this molecule is maybe produced at a very low titer, are we planning to intensively grow this organism? Is there any current research about in vitro biosynthesis in model hosts for production at industrial scale? Please bring, more discussion and analysis about the significance of marine natural products as anti-candidal agents.
Answer: As suggested by reviewer an appropriate modification was included in the revised manuscript.

Reviewer 2 Report
This review manuscript provides a thorough summary of anti-candidal substances derived from marine organisms. While previous reviews have explored the antifungal activity of marine natural products, none have specifically focused on candidiasis. Given the medical importance of candidiasis, this review provides important insights for relevant researchers. The manuscript is well-structured and supported by numerous references. The tables present clear organization of compound names, origins, activities, and literature information.
However, there are a few points that require revision to enhance the clarity and consistency of the manuscript. As the review primarily focuses on secondary metabolites, the inclusion of chemical structures is crucial for a comprehensive understanding. Currently, the drawing format of the compounds lacks consistency. In addition, the compounds are not numbered, which may hinder effective comparison with the text. To address this, I suggest implementing the following revisions:
1: All compounds described in the text should be drawn with the same format.
2: Assign consecutive numbers to each compound and ensure that the corresponding number is clearly indicated in the text.
3: The title of the table is incorrect; they are not Phytochemical.
By implementing these revisions, the manuscript will significantly improve its visual clarity and overall coherence.
Author Response
1. All compounds described in the text should be drawn with the same format.
Answer: In this review we were included around 293 compounds, the structure of some compounds was provided in the Figures. Considering the concern raised by the reviewer while including these structures the number of figures will be exceeded than specified by the JoF. That’s why we have restricted ourselves to included only few structures of the compounds.
2. Assign consecutive numbers to each compound and ensure that the corresponding number is clearly indicated in the text.
Answer: All the compounds discussed in this manuscript were consecutively numbered to meet requirement of the readership.
3. The title of the table is incorrect; they are not Phytochemical.
Answer: Title of the table was modified and presented appropriately.

Round 2
Reviewer 1 Report
This review paper entitled “Anti-Candidal Marine Natural Products: A Review” submitted by Arumugam and all., presents marine natural products identified as active compounds against Candida sp. pathogens, especially C. albicans. For this review, authors have classified molecules with anti-candida activity based on their origin (Sponge, Algae,...). All data have been summarized as tables that include source, target and MIC values. In this manuscript, authors discuss potential of marina natural products for discovery of anti-candida agents, detail current and future challenges and provide directions for future investigations. After a first revision, I have noticed that all my comments and questions have been taken into consideration by the authors. Consequentially, I consider that this manuscript can be accepted for publication after a final proofreading.
Final proofreading for minor typos and grammar before acceptance for publication.
Author Response
As suggested by the reviewer this version of the manuscript was carefully proofread for their typo and grammatical errors.